# Nose-to-Heart Approach: Unveiling an Alternative Route of Acute Treatment

**DOI:** 10.3390/biomedicines12010198

**Published:** 2024-01-16

**Authors:** Paraskevi Papakyriakopoulou, Georgia Valsami, Nikolaos P. E. Kadoglou

**Affiliations:** 1Laboratory of Biopharmaceutics and Pharmacokinetics, Section of Pharmaceutical Technology, Department of Pharmacy, School of Health Sciences, National and Kapodistrian University of Athens, 15771 Athens, Greece; ppapakyr@pharm.uoa.gr (P.P.); valsami@pharm.uoa.gr (G.V.); 2Medical School, University of Cyprus, Nicosia 2029, Cyprus

**Keywords:** nose-to-heart delivery, acute myocardial infraction, angina pectoris, hypertensive crises, cardiac arrythmias, acute treatment

## Abstract

Intranasal (IN) administration has emerged as a novel approach for rapid systemic absorption, with potential applicability in the management of acute cardiovascular events. This review explores the evolution of IN cardiovascular pharmacotherapy, emphasizing its potential in achieving systemic effects and bypassing the first-pass metabolism associated with oral administration. The extensive vascularization of nasal mucosa and a porous endothelial basement membrane facilitate efficient drug absorption into the bloodstream. The IN route ensures a critical swift onset of action, which allows self-administration in at-home settings. For instance, etripamil nasal spray, a first-in-class formulation, exemplifies the therapeutic potential of this approach in the treatment of spontaneous supraventricular tachycardia. The review critically assesses studies on IN formulations for angina, acute myocardial infarction, hypertensive episodes, and cardiac arrhythmias. Preclinical evaluations of beta-blockers, calcium-channel blockers, and antianginal drugs demonstrate the feasibility of IN administration for acute cardiovascular events. A small number of clinical trials have revealed promising results, emphasizing the superiority of IN drug delivery over oral administration in terms of bioavailability and onset of action. Unambiguously, the limited clinical trials and patient enrollment pose challenges in generalizing experimental outcomes. However, the nose-to-heart approach has clinical potential.

## 1. Introduction

Intranasal (IN) administration of cardiovascular pharmaceutical agents was first proposed in 1994 by Landau and his colleagues [1]. However, sparse studies on the development of IN cardiovascular pharmacotherapy are reported in literature. Due to minimal scientific interest and progress in the following decades, this way of administration remained largely unexplored. IN delivery still represents an emerging and innovative approach to achieve systemic effects of medications in cardiovascular events requiring acute therapeutic intervention, such as acute myocardial infarction, hypertensive crises, and cardiac arrhythmias [2,3,4]. The advantage of this non-invasive route is the rapid drug absorption through the nasal mucosa and the prompt systemic circulation, since it bypasses the first-pass metabolism and/or gastrointestinal breakdown that interferes with oral administration [1]. The nasal mucosa is characterized by extensive vascularization with a relatively broad absorption area, because of a porous and thin nasal endothelial basement membrane [5]. All the aforementioned properties of nasal mucosa, leading to high permeability, enable more efficient and faster systemic drug absorption, even in small amounts, than the oral route [6]. IN delivery could be considered as equivalent to the intravenous mode of administration [7,8]. Additionally, IN administration has the potential of self-administration in at-home settings, thus avoiding continuous medical supervision. Self-administration of nasal delivery is easy and empowers individuals to control their emergent medical needs, such as spontaneous paroxysmal supraventricular tachycardia (SVT) [9].

From the clinical perspective, etripamil nasal spray represents the first-in-class formulation for IN administration of cardiovascular drugs. It is a pioneering pharmaceutical product specifically designed for the treatment of SVT and is currently undergoing evaluation [10]. Recently, Hassan et al. (2023) [4] investigated the potential of IN delivery systems for hypertension management. Moreover, Wang et al. (2023) [11] tested the potential of IN extracellular vesicles combined with hydrogel to treat myocardial ischemia–reperfusion injury through the inhibition of inflammation and protection of endothelial function. In the past, there was intensive research into the development of IN formulations and their clinical application in emergency treatment of several cardiovascular conditions, like acute myocardial infarction and angina pectoris [12,13]. The administration of IN solutions of aspirin [14] and nitroglycerin [15] was proposed as an alternative and effective route. Despite the reported advantages of IN, over the previous decade scientific interest was not sustained and the research for the clinical application of IN faded. Some of those drugs tested in the past are not used anymore, or their indications have been withdrawn. However, in recent years, IN administration has rekindled interest, in the context of emergency treatment, due to the aforementioned advantages. Large ongoing research projects aim to introduce this way of administration into cardiovascular disease management.

The objective of this comprehensive review is to provide a concise and critical analysis of the nose-to-heart approach in the acute treatment of emergency cardiovascular events, such as heart attacks, tachycardia, and hypertensive crises. Furthermore, we aimed to elucidate the therapeutic potential of this delivery alternative based on findings from experimental and clinical studies.

## 2. Methods: Literature Search

A thorough investigation was conducted across electronic databases such as PubMed, Scopus, and Google Scholar to identify English-language publications. An extended description of the nose-to-heart approach is presented, based on the anatomy and physiology of systemic pathways via IN administration. The search covered all publications from 1980 to 2023, serving as a comprehensive reference for the development of IN formulations for angina pectoris, acute myocardial infraction, hypertensive crises, and cardiac arrhythmias, including also the preclinical and clinical studies of nose-to-heart delivery for these cardiovascular diseases. The search criteria included terms such as nose-to-heart delivery, IN systemic delivery, alone or combined with the names of the drugs and the diseases. The authors excluded studies lacking full text, those in languages other than English, and conference abstracts. After screening titles and abstracts, we incorporated into the review 20 full-text preclinical and clinical studies.

## 3. Nose-to-Heart Approach

This section provides a concise description of the experimental results and a thorough interpretation.

### 3.1. Anatomical and Physiological Characteristics of Systemic Absorption via IN Administration

Oral administration remains the most conventional non-invasive route for systemic drug delivery which ensures higher compliance, especially among patients with chronic diseases. However, the limited bioavailability of certain compounds, due to inadequate absorption or extensive metabolism along the gastrointestinal tract, has shifted interest towards alternative administration methods. IN drug delivery has gained significant recognition as a viable and versatile substitute to the oral route [16]. A comprehensive understanding of nasal cavity anatomy and physiology is essential for grasping the principles governing nose-to-heart delivery (NHD). Historically, nasal administration was primarily used for symptomatic relief, prophylaxis, or treatment of local nasal diseases. However, in recent years, the nasal mucosa has evolved into a promising route for systemic and nose-to-brain drug delivery, which takes advantage of its unique structural, physiological, and histological features [17]. The nasal cavity’s total area is approximately 150–160 cm^2^, with a volume capacity of 15 to 20 mL [18]. Featuring cilia and glands, its internal surface acts as a protective barrier against dust, pathogens, and particles, with each nostril further divided into four areas: nasal vestibule, atrium, respiratory region, and olfactory area [19].

Particularly, the nasal cavity compromises two main areas, the olfactory and respiratory, in a surface ratio of approximately 1:9. The olfactory area is mainly responsible for the drug transfer to the brain, while the high vascularization of the respiratory region allows for rapid and extensive drug absorption into the bloodstream bypassing first-pass metabolism [20]. The respiratory region comprises superior, middle, and inferior turbinates [21]. The respiratory nasal mucosa’s lamina propria, deeper than the columnar epithelium, includes blood and lymphatic vessels, nerves, glands, and immune cells [22]. Goblet cells secrete mucin, forming a double mucus layer, modulating the nasal mucosa’s overall pH (5.2 to 8.1) [23]. Due to its large, highly perfused surface area, the respiratory region could be a great target for systemic drug delivery following nasal administration [16]. The trigeminal neurons and vasculature make it a significant site for nose-to-brain delivery, albeit with slower transport compared to the olfactory nerve [24].

The mechanisms governing drug delivery through the nasal mucosa encompass both paracellular and transcellular pathways. Notably, the paracellular route is restricted by the molecular weight of water-soluble compounds, with absorption diminishing as molecular weight surpasses 1000 Daltons [25]. The low-molecular-weight drugs administered via the nasal route usually show high bioavailability and low variability, while the opposite happens for high-molecular-weight drugs [8]. Conversely, in the transcellular route, the drug’s lipophilicity influences transport rates [26,27]. The respiratory region, characterized by a ciliated pseudostratified columnar epithelium and a rich vascular network, emerges as a key player in drug absorption into the bloodstream. Drug molecules dissolved into nasal fluids traverse the respiratory epithelium, composed of distinct layers with unique properties [28]. Lipophilic drugs diffuse through lipid bilayers, while hydrophilic drugs may exploit specific transporters on cell surfaces. Beyond direct absorption, some drugs enter the lymphatic system, with nasal cavity lymphatic drainage transporting drugs to regional lymph nodes and ultimately contributing to systemic distribution [29].

### 3.2. Nose-to-Heart Delivery

The rich blood supply of the nasal mucosa is fundamental for the success of systemic drug delivery [5]. Arterial blood supply originates from several major arteries, primarily the ophthalmic, sphenopalatine, and facial arteries (Figure 1). These arteries irrigate a dense capillary network and large venous sinusoids near the turbinate respiratory zone. Venous return involves the sphenopalatine, facial, and ophthalmic veins, ultimately draining into the internal jugular veins and proceeding down to the right heart through the superior vena cava [30]. These anatomical characteristics contribute to the fast onset of action. Furthermore, the velocity of blood flow significantly impacts the systemic nasal absorption of drugs, maintaining a concentration gradient from the absorption site to the blood, crucial for diffusion-driven drug absorption. Consequently, alterations in local vascular homeostasis, influenced by pathological conditions and/or drugs, can profoundly affect the rate and extent of the systemic absorption of intranasally administered drugs, necessitating careful assessment [16]. Hence, the high density of the vasculature of the nasal mucosa, in conjunction with its extended surface area, the high permeability, and basal membrane porosity, render the nasal administration route highly appealing for a diverse range of therapeutic compounds for acute cardiovascular treatment [5].

## 4. Cardiovascular Diseases

From a clinical perspective, the IN route in acute treatment of cardiovascular events constitutes a feasible, non-invasive, effective alternative, because of the more rapid onset of drug action compared to other modes of parental administration such as sublingual or transdermal. In those episodes, time matters, determining the final result (Figure 1). In an attempt to elucidate the potential clinical application of NHD, we extensively investigated literature for the development of nasal formulations of antianginal drugs, beta-blockers, calcium-channel blockers (CCBs), and their pharmacokinetic or pharmacodynamic patterns in animal models. These preclinical and clinical studies are summarized in Table 1 and Table 2. (For Table 1 and Table 2, see the end of Section 4).

### 4.1. Angina Pectoris and Acute Myocardial Infraction

Angina pectoris manifests as precordial discomfort or pressure, arising from transient myocardial ischemia without causing myocardial necrosis [31]. Acute myocardial infraction (AMI), commonly known as heart attack, is a potentially lethal event which results from sudden and persistent severe reduction in or complete cessation of blood flow to a portion of the heart muscle, leading to local necrosis [32]. In acute treatment, aspirin can ameliorate platelet aggregation and limit coronary artery obstruction with profound benefits for AMI progression and prognosis [33]. For those reasons, aspirin remains the cornerstone of treatment for acute coronary syndromes (ACS) and chronic coronary syndromes (CCS) [34]. In the acute setting of ACS, i.v. or sublingual administration of nitrates may considerably relieve chest pain. In CCS, the invasive coronary revascularization is the first-line therapy for ischemic chest pain. However, nitrates are highly recommended in therapy for acute effort angina or for angina prophylaxis [35]. Pharmacological therapy and prevention of angina is not uncommon in clinical practice, since a significant proportion of patients suffer from coronary microvascular disease or refractory angina. Nitroglycerin is a well-known member of nitrates with vasodilator action, enhancing coronary blood flow and relieving angina [1,36].

Regarding the time-sensitive nature of AMI, IN drug administration could contribute to time saving. Aspirin absorption through the nasal cavity was evaluated using a rat model by Hussain et al. (1992) [14] with a 2 mg IN administration, reporting the maximum concentration at 30 min and a 100% bioavailability. A similar T_max_ value was reported for acetylsalicylic acid (T_max_ = 27 ± 8 min) in healthy volunteers receiving 162 mg of chewable low-dose aspirin. For the attainment of salicylic acid maximum concentration, more time was required, resulting in a T_max_ of 69 ± 21 min [37]. AMI management with acute intervention and shorter timeframes would certainly be desirable, especially if it could occur within a few minutes. In this context, nitroglycerin is mostly administered sublingually either in the form of a tablet or spray at doses of 0.3 to 0.6 mg every 5 min. Sublingual nitroglycerin tablets [Nitrostat^®^ (Nitroglycerin Sublingual Tablets, USP)] can lead to a peak plasma concentration 6 to 7 min post-dose [38]. Nitroglycerin can also be administered topically as an ointment or transdermal patch, as well as intravenously in the emergency department, at a rate of 5 to 10 μg/min, gradually titrating up to a maximum of 400 μg/min, when the symptoms persist [39]. IN administration of nitroglycerin has been reported since 1981 by Hill et al. [15] as an alternative to topical or sublingual delivery in the case of myocardial ischemia or a significant elevation of pulmonary artery and diastolic pressure by reducing preload and afterload. This study revealed the PKs of 0.8 mg nitroglycerin IN solution from half a minute to 32 min after administration. The PK profiles of central, arterial, and peripheral blood demonstrate rapid absorption within 1–2 min, resulting in higher levels compared to sublingual administration [15]. This is an old study which requires further validation.

β-blockers play a significant role as anti-ischemic drugs in patients with CCS, while they improve prognosis after myocardial infarction or in patients with heart failure [34,35,40]. Moreover, β-blockers have been recommended for acute treatment and chronic prevention of ventricular and supra-ventricular arrhythmias [41]. Metoprolol, a member of the β-blockers family, has been assessed at the preclinical level as a candidate medication for angina pectoris and tachyarrhythmias [13]. This scenario was simulated in rabbits using isoprenaline to induce tachycardia. The animals were intranasally treated either with the drug solution or with sodium alginate microspheres at the dose of 2 mg/kg. The nasal solution exhibited maximum efficacy 10 min after administration, making it a suitable option for acute treatment. In contrast, the microspheres demonstrated sustained release of the substance, with higher inhibition of isoprenaline-induced heart rate being observed 180 to 300 min after administration [13]. Previously, Kilian and Müller documented the significance of formulation viscosity on contact time with the mucosa and, consequently, on achieved bioavailability. The incorporation of 2% methylcellulose into the synthesis increased serum exposure. However, the addition of 0.1% polysorbate-80 as absorption enhancer reduced bioavailability, probably due to the high drug entrapment [12]. In a clinical context, the IN administration of both metoprolol and alprenolol in methylcellulose was performed on volunteers without defining their medical history. A comparison of the IN route with oral and sublingual administration indicated that the compound’s hydrophobicity is critical for nasal absorption. Specifically, poor absorption was observed for the hydrophilic metoprolol, whereas the lipophilic alprenolol showed improved bioavailability after nasal delivery [42].

Another β-blocker, carvedilol comprises an essential drug for acute coronary syndromes and chronic heart failure, improving symptoms and prognosis [43,44]. A controlled release profile was also observed in the case of carvedilol microspheres administered intranasally, in radiolabeled form, at the dose of 3 mg/kg in rabbits. The maximum radioactivity was observed 1 h after administration [45]. However, the results of this study should be critically considered in terms of NHD, as the gamma scintigraphy evaluation revealed a relatively minor signal in heart tissue, which was not commented on by the authors [46]. The first-generation, non-selective β-blocker propranolol was primarily indicated for hypertension and angina pectoris. Taking advantage of its rapid absorption across the nasal mucosa, Hussain et al. (1980) [47] demonstrated the equivalence of nasal delivery to the intravenous administration, while Landau et al. (1993) [48] found that a 5 mg dose of propranolol nasal spray 15 min before exercise can promptly block beta receptors, enhancing exercise tolerance in 16 patients with effort-induced angina, in the context of a blinded, randomized, crossover design study. Nowadays, selective b1 receptor blockers have prevailed, and the usage of propranolol has been limited to acute therapy of symptomatic ventricular or supra-ventricular extra-systoles, regarding its shorter action compared to other β-blockers. It would be interesting to test in the future the IN administration of propranolol for acute therapy of those arrhythmias.

The temporarily restriction of blood supply (ischemia) to the heart tissue and its subsequent restoration (reperfusion) can lead to an inflammatory response and cellular dysfunction or death, commonly referred to as myocardial ischemia–reperfusion injury [49]. Among the various signaling pathways implicated in endothelial function, the guanylate cyclase (GC) pathway plays a crucial role in maintaining vascular homeostasis. Disruptions in this pathway can have profound implications, as GC is responsible for catalyzing the conversion of guanosine triphosphate (GTP) to cyclic guanosine monophosphate (cGMP) [50]. Additionally, nitric oxide (NO), a key regulator of vascular tone and a potent vasodilator, stimulates GC to produce cGMP. Reduced bioavailability of NO, often associated with oxidative stress, inflammation, or other pathological conditions, can lead to decreased guanylate cyclase activity and subsequent cGMP synthesis [51].

Addressing myocardial ischemia–reperfusion injury during the acute stage is often associated with a systemic inflammatory response, potentially hindering the restoration of cardiac function. In a recent study of Wang et al. (2023) [11] extracellular vesicles (EVs) combined with a two-component supramolecular hydrogel were intranasally administered in a myocardial I/R injury mouse model. The administration method and formulation allowed for efficient delivery of EVs into the peripheral blood, suppressing the inflammatory response, safeguarding endothelial cells in mice experiencing myocardial ischemia–reperfusion injury. Enhanced cardiac function and smaller infarct size were observed, revealing the potential of this therapeutic approach in the treatment of cardiac ischemia–reperfusion injuries [11].

### 4.2. Hypertensive Crisis

Hypertensive crises (HC) are characterized by pronounced surges in blood pressure, accompanied by vital organ injuries. Vascular endothelial injury evolves, leading to a cascade of detrimental effects if not promptly treated [52]. HC most often occur in individuals with pre-existing hypertension with irritated systemic vascular resistance and a failure of cerebral blood flow autoregulation [53]. Acute treatment of HC is necessary, but it should avoid an exaggerated and steep blood pressure decline. Therefore, the onset of drug action should be rapid and steadily controlled throughout. Up to now, the effective acute therapy of HC is still the subject of investigation and highly depends on the underlying condition. The previously used CCBs were associated with detrimental rapid blood pressure lowering [54]. However, nifedipine, a CCB of the dihydropyridine subclass, remains the first-line agent for women with pre-eclampsia [55].The IN administration of nifedipine was evaluated by Kubota et al. (2001) [56] in healthy volunteers for acute blood pressure lowering at a dose of 8.12 ± 0.32 mg. The study showed that IN administration exhibits the most favorable early increase in serum concentration and effectively reduces blood pressure compared to oral or sublingual routes. Similar findings were reported in the case of another dihydropyridine CCB, nicardipine, which was found to produce a better hemodynamic impact than oral and tracheal delivery [2,57]. The main challenges of nicardipine IN solution were the dose inconsistency and low bioavailability as a dose fraction flowing into the gastrointestinal tract via the internal nares and the pharynx [56]. These constitute common issues in nasal delivery, highlighting the need for pioneering formulations with targeted deposition and more efficient absorption into the nasal cavity [58]. Considering that clevidipine is endorsed for blood pressure control in patients with acute ischemic stroke, it seems that CCBs may be recommended in specific hypertensive states (e.g., acute aortic syndromes). If the intensive short-acting formulation is avoided, CCBs still have the potential to regain their place in therapeutic algorithms of hypertensive peak or crisis [59].

Historically, beta-blockers were among the most prescribed medication for first-line treatment of hypertension, but their usage has been limited nowadays [60]. Among β-blockers, carvedilol, timolol, and propranolol have been assessed preclinically either in the form of a nasal solution or in more complex formulations [2,61,62,63]. The study of Kar and Singh (2023) [61] applied design of experiments (DoE) methodology to develop and optimize carvedilol-loaded cationic nanoliposomes in the form of in situ nasal gel, using P90H and DOTAP Cl (cationic) lipids. The comparative pharmacokinetic study of carvedilol oral suspension and oral cationic liposomes, and the liposomes formulated into in situ nasal gel, revealed the superiority of the nasal formulation. Specifically, an almost seven-fold increase in drug C_max_ in rabbit plasma and 1 h earlier T_max_ were observed for the in situ gel compared to the pure drug and the orally administered liposomes. Additionally, a five-fold greater relative bioavailability was calculated compared to oral carvedilol suspension [61]. Similarly, Jagdale et al. (2016) [62] applied DoE for the formulation of timolol maleate in a thermoreversible in situ gel for the treatment of hypertension. HPMC and Poloxamer 407 were used as independent variables of a factorial design, and the prepared formulations were characterized in in vitro and ex vivo experiments. The optimized formulations have significant outcomes in terms of gelation temperature, mucoadhesive strength, and drug release, as well as high drug release with a sustained profile [62]. However, currently timolol is primarily used for the treatment of elevated intraocular pressure or open-angle glaucoma [64], and propranolol is prescribed for tachyarrhythmias in children or adolescents or young adults [65,66].

### 4.3. Cardiac Arrhythmias

Symptomatic, paroxysmal, cardiac arrhythmias are abnormal heart rhythms derived from several cardiac abnormalities (dysregulation of electrical conduction system, enhanced automaticity, structural heart disease) [67], or common triggers such as viral illness, alcohol, exercise, caffeine, anxiety, and recreational drugs [68,69]. Irregular rhythms may present either as fast (tachyarrhythmias) or slow (bradyarrhythmias) heart rates [70]. Many cardiac arrhythmias become more frequent with advancing age, while the treatment options include interventional procedures (ablation, pacemaker implantation maze procedure) or daily oral antiarrhythmics [71]. Additionally, vagal maneuvers, intravenous antiarrhythmics, and synchronized cardioversion can be employed for acute treatment of tachyarrhythmias [72]. Oral beta-blockers and non-dihydropyridines CCBs can be self-administered as a “pill-in-the-pocket” approach for acute treatment of repeated episodes of recognized arrhythmias with an average time of 30 min or more [73]. However, the effectiveness of this therapeutic strategy is questioned and faces significant risks [74]. Oral propranolol has been suggested for supraventricular tachycardias in children and infants [66], while no reports are available for IN administration. Only two formulation approaches have been published for propranolol, in the form of nasal spray, developed and evaluated by Landau et al. (1993) [48], as well as in mucoadhesive microspheres using gelatin A and chitosan. The microspheres were evaluated applying in vitro techniques that revealed a particle size of 1–50 μm, good mucoadhesive properties, and a sustained drug release profile [48].

Verapamil is a CCB which has emerged as a significant therapeutic option for the management of SVT [75]. Oral verapamil has been suggested as a second-line therapy for rate control in atrial fibrillation [76], cardiac extra-systolic arrhythmia alone or combined with class IC or III antiarrhythmic drugs [77]. The efficacy of IN verapamil has been tested in a dog model, while a pharmacokinetic study in rabbits revealed that the IN delivery leads to higher bioavailability compared to the oral route [72,78]. However, IN verapamil in humans resulted in almost 16.1% absolute bioavailability, attributed to dose ingestion through the nasopharynx [72,79].

Recently, an etripamil nasal spray was developed for self-administration in the case of SVT. Etripamil is a novel non-dihydropyridine calcium channel blocker which is currently undergoing phase III clinical trials for the treatment of SVT. This drug is characterized by a short half-life of 20 min, in contrast to those of the other members of the CCBs, which range from 2 to 7 h (verapamil: 3–7 h, diltiazem 2–7 h), and T_max_ of 8 min [3,80]. These properties render it an ideal candidate for non-invasive acute treatment in SVT, confirmed also in preclinical studies in which a quick distribution, within 15 min, was observed [3]. The alternative drug is adenosine, which is always intravenously administered under monitoring at hospital, due to its potential side effects [81]. Etripamil nasal spray was assessed for its efficacy and tolerability in the context of phase 1 and 2 of clinical trials, resulting in a conversion rate of 95% versus the 35% achieved in the placebo group [82]. The mean conversions time was less than 3 min, with a range from 2 to 30 min. The dose of 70 mg was found to be the optimum to ensure both efficacy and safety. All the patients presented at least one side effect, mainly related to the mode of administration, such as nasal discomfort and congestion, rhinorrhea, and oropharyngeal pain. This tolerability profile was not found to be significantly different from the placebo [74]. A phase III study was performed for the dose of 70 mg. The self-administration of etripamil nasal spray in adults experiencing symptomatic, sustained SVT (NODE-301 study) was well-tolerated when utilized in a medically unsupervised setting outside the healthcare environment. A STV conversion rate of 53.7% was observed at the 30-min time point, whereas 34.7% was observed in the placebo group [10]. As part of the NODE-301 study, repeated administration of the treatment was assessed in 104 patients with SVT and persistent symptoms and found to be more effective than both placebo and the one-dose treatment [83]. The three above-mentioned studies, which included 466 participants in total, was systematically reviewed, and a meta-analysis was performed by Abuelazm et al. (2023) [9], who concluded that etripamil nasal spray was effective in SVT within 60 min, leading to a reduced need for medical supervision and intervention, as well as emergency department visits along with good tolerance, with some mild local discomfort [9]. The efficacy of etripamil nasal spray has been further evaluated in cases of repeated episodes of SVT at the dose of 70 mg and found to be safe with no reports of syncope, hypotension, atrioventricular block, or bradycardia. Based on this extension study, the nasal spray is considered well tolerated, allowing for the self-administration of the drug without medical supervision [84].

**Table 1 biomedicines-12-00198-t001:** Formulation and in vivo preclinical studies based on nose-to-heart delivery approach, listed by year.

Drug	Dose	Animal Model	Disease	Main Outcomes	Year	Reference
verapamil	0.75 mg/kg	dog	rate control/cardiac arrythmias	higher bioavailability compared to the oral route	1985	[77]
aspirin	2 mg	rat	acute myocardial infraction	T_max_ at 30 min, 100% bioavailability	1992	[14]
metoprolol	1 mg	rat	angina pectoris	2% methylcellulose increases the formulation viscosity and the contact time with nasal mucosa	1998	[12]
metoprolol	2 mg/kg	rabbit	angina pectoris	maximum efficacy of drug solution at 10 min, sustained release microspheres with higher inhibition observed at 180 to 300 min	2003	[13]
propranolol	−	Formulation study	−	microspheres of gelatin A and chitosan, of 1–50 μm, with good mucoadhesive properties and sustained release profile	2007	[63]
carvedilol	3 mg/kg	rabbit	unstable angina	maximum radioactivity observed 1 h after administration	2010	[45]
carvedilol	1 mg/kg	rabbit	Hypertensive crises	almost 7-fold increase in drug C_max_ in rabbit plasma with the in situ gel, and 1 h earlier T_max_ compared to pure drug and oral liposomes	2023	[61]

C_max_, the highest concentration of a drug in the blood; T_max_, time to peak drug concentration.

**Table 2 biomedicines-12-00198-t002:** Clinical studies based on nose-to-heart delivery approach, listed by year.

Drug	Dose	Disease	Main Outcomes	Year	Reference
propranolol	10 mg	angina pectoris	equivalence of nasal delivery to the intravenous administration	1980	[48]
nitroglycerin	0.8 mg	acute myocardial infraction	T_max_ at 1–2 min, higher levels compared to sublingual administration	1981	[15]
metoprolol	20 mg	angina pectoris	poor nasal absorption of methylcellulose nasal solution due to high hydrophilicity	1986	[42]
alprenolol	10 mg	hypertension	improved bioavailability of methylcellulose nasal solution due to high lipophilicity	1986	[42]
nicardipine	N/A	Hypertensive crises	better hemodynamic impact of IN administration than oral and tracheal delivery	1990	[57]
propranolol	5 mg	angina pectoris	nasal spray 15 min before exercise enhances exercise tolerance	1993	[48]
verapamil	5 mg	cardiac arrythmias	low absolute availability after IN administration, almost equal to 16.1%	1993	[78]
nifedipine	8.12 ± 0.32 mg	hypertensive crises	early increase in serum concentration and reduced blood pressure compared to oral or sublingual routes	2001	[56]
etripamil	70 mg	SVT	Phase I and II: safe and efficacious dose	2018	[73]
etripamil	70 mg	SVT	Phase I and II: conversion rate of 95% versus 35% achieved in placebo group	2020	[82]
etripamil	70 mg	SVT	Phase III: well-tolerated, conversion rate of 53.7% at 30 min versus 34.7% observed in placebo group	2022	[10]
etripamil	70 mg	SVT	Phase III: more effective than placebo; repeated dose was more effective than one-dose treatment	2023	[83]

C_max_, the highest concentration of a drug in the blood; SVT; Supraventricular Tachycardia; T_max_, time to peak drug concentration.

## 5. Disadvantages of IN Drug Delivery

IN administration has emerged as a promising avenue for drug delivery to the heart, offering a non-invasive and potentially efficient means of bypassing traditional routes. This method is particularly appealing for certain cardiovascular medications, given its potential to enhance bioavailability and reduce the need for invasive procedures. However, it is imperative to acknowledge that the application of IN administration in cardiac drug delivery is not without its challenges. Drawbacks such as variable absorption rates, irreversible damage of the nasal mucosa, and the limited volume of drug that can be delivered pose considerations that necessitate careful evaluation. This administration method may cause irritation and discomfort, leading to symptoms such as sneezing, itching, or a burning sensation, especially when used for chronic diseases requiring daily dosing. Nasal dryness and increased susceptibility to infections are also concerns [6]. The impact and occurrence of these issues lessen in the cases of acute treatment requiring an instant/sole administration for the management of the situation. In this case, a convenient and minimum-steps-to-assemble nasal device constitutes the main point of concern to achieve an effective treatment [28].

The success of IN drug delivery and its targeted impact on certain organs relies not only on nasal cavity and epithelial conditions but also on the type of formulations and devices employed. While conventional nasal formulations in the market typically consist of simple solutions or suspensions, emerging evidence suggests the necessity for advanced approaches to usher in the next generation of nasal products [85]. Special attention is required for the rapid drug elimination and the dosing inconsistencies that can occur due to interindividual variability of the nasal mucosa environment or improper administration [6].

Additionally, stability issues of liquid formulations, as well as inadvertent drug inhalation into the lungs, are common challenges. The selection of formulation type and excipients is guided by factors such as the solubility and stability of the active drug, as well as the required concentration to deliver an effective dose within a typical range of from 25 to 250 μL of spray [86]. Moreover, the nasal route, while promising rapid drug absorption and bypassing first-pass metabolism, is inherently associated with the risk of unintentional pulmonary exposure. One primary challenge lies in the intricate anatomy of the nasal cavity, which shares a common pathway with the respiratory system. The dynamic interplay between nasal airflow, particle size, and deposition patterns poses a substantial obstacle. Ensuring that the administered drug remains confined to the nasal mucosa and is not carried into the lower respiratory tract requires precise control over formulation characteristics [87]. Strategies such as particle engineering, optimization of droplet size, and innovative delivery devices must be employed to minimize the risk of pulmonary deposition. Particularly, to avoid potential undesired effects from the lower respiratory tract due to inhalation in lungs, droplet diameters after spraying should exceed 10 μm, producing only a low proportion of smaller droplets (<10 μm) [88]. Additionally, factors influencing mucociliary clearance, the presence of protective mucus barriers, and variations in nasal physiology among individuals further complicate the task [89]. Addressing these challenges is crucial for developing nasal drug delivery systems that achieve targeted systemic effects without compromising the safety of other organs such as lungs and the brain.

## 6. Conclusions

This comprehensive review aims to summarize and critically discuss the nose-to-heart approach in the acute treatment of emergency cardiovascular events, such as heart attack, tachyarrhythmias, and hypertensive crises, based on findings from experimental and clinical studies. Due to the growing number of studies in past years, the superiority of IN drug administration in acute cardiac treatment—regarding fast onset of action and enhanced systemic bioavailability compared to the oral route—is well documented in preclinical animal studies. However, the extrapolation of experimental results to clinical practice should be carried out with caution. There is a limited number of old clinical trials, with small samples confined to a limited number of drugs, some of which do not remain as first-line therapy or whose indications have been withdrawn. Currently, there is a renewed scientific interest in the IN approach, and acute therapy with the new drugs should be updated. Despite the promising preclinical results and the high and immediate bioavailability of IN-administered drugs, there are significant limitations for the clinical assessment of this alternative approach. Additionally, well-designed randomized clinical trials are imperative in order to elucidate the therapeutic potential of IN delivery and draw robust conclusions for first-line medications.

## Figures and Tables

**Figure 1 biomedicines-12-00198-f001:**
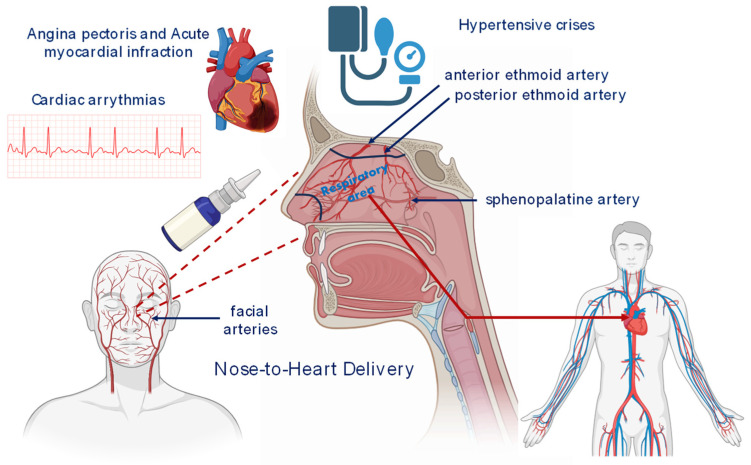
Nose-to-heart delivery approach, mediated via the contribution of anterior and posterior ethmoid, as well as the sphenopalatine arteries, showcasing potential applications in acute myocardial infarction, angina pectoris, hypertensive crises, and cardiac arrhythmias.

## Data Availability

Data sharing not applicable.

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
