# Peer review of "Nose-to-Heart Approach: Unveiling an Alternative Route of Acute Treatment"

_biomedicines, 2024, doi:10.3390/biomedicines12010198_

Round 1
Reviewer 1 Report
Comments and Suggestions for Authors
The authors undertook a very difficult task. They prepared a review on intranasal drug administration in various cardiac conditions. interesting manuscript, but unfortunately in its current form it is misleading. The authors used a number of publications from the last 30 years. Research that suggested the use of some drugs at that time is not necessarily current (e.g. verapamil is not used in the prevention of AF events).
I believe that in order to be able to think about publication, it is necessary to:
- each subchapter regarding clinical indications must begin with a few-sentence introduction describing the action in a given indication COMPLIANT with the ESC/EHRA/ACC guidelines!
- historical indications, not used for years, can only be discussed in terms of the pharmacokinetics of absorption of the drug administered intranasally, but therapeutic use cannot be suggested.
- in the introduction, please discuss in which indications the use of IN may make sense. IMHO, this is primarily a treatment for patients with arrhythmias, but not all of them, only those who cannot be treated with ablation or ablation attempts have been unsuccessful. Ablation is the first choice method in most cases. In other indications - for example, nifedipine - has not been used for emergency treatment of increased blood pressure for many years - too high risk of cardiovascular events in populations. The exception may be pregnant women and people with narrowing of the renal arteries (but at the same time with healthy coronary and cerebral arteries). In this way, please analyze all indications. Antianginal drugs are used very rarely due to wide access to invasive treatment. This procedure always has priority.
- in the light of the above, it is worth thinking about changing the title... it is difficult to call it a "gold player"...
- please provide a broader discussion of the mentioned endothelial dysfunction, primarily through the guanylate cyclase pathway.
I think the topic is interesting (therefore worth publishing). The work cannot mislead the reader and therefore requires expansion and thorough reconstruction of the chapters, taking into account current clinical recommendations.
Author Response
Reviewer 1
The authors undertook a very difficult task. They prepared a review on intranasal drug administration in various cardiac conditions. interesting manuscript, but unfortunately in its current form it is misleading. The authors used a number of publications from the last 30 years. Research that suggested the use of some drugs at that time is not necessarily current (e.g. verapamil is not used in the prevention of AF events). I believe that in order to be able to think about publication, it is necessary to:
- each subchapter regarding clinical indications must begin with a few-sentence introduction describing the action in a given indication COMPLIANT with the ESC/EHRA/ACC guidelines!
We really appreciate the reviewer’s suggestion. Indeed, most clinical data derived from old small studies, using first generation drugs or those less indicated anymore. For instance, verapamil is a second line drug for rate control in AF which remains the preferred strategy in elderly patients, where the safety and efficacy of anti-arrhythmic medications is under investigation according to recent ESC guidelines (2020). We have outlined this drawback and we will further emphasize that this route of administration was tested in the past, its use has been fainted, but we believe that it can regain scientific interest and clinical application. We have changed the structure providing a few sentences introduction of each drug and its indications.
- historical indications, not used for years, can only be discussed in terms of the pharmacokinetics of absorption of the drug administered intranasally, but therapeutic use cannot be suggested.
Following reviewer’s suggestion, we have removed references of drugs which are not recommended anymore, like timolol.
- in the introduction, please discuss in which indications the use of IN may make sense. IMHO, this is primarily a treatment for patients with arrhythmias, but not all of them, only those who cannot be treated with ablation or ablation attempts have been unsuccessful. Ablation is the first choice method in most cases. In other indications - for example, nifedipine - has not been used for emergency treatment of increased blood pressure for many years - too high risk of cardiovascular events in populations. The exception may be pregnant women and people with narrowing of the renal arteries (but at the same time with healthy coronary and cerebral arteries). In this way, please analyze all indications. Antianginal drugs are used very rarely due to wide access to invasive treatment. This procedure always has priority.
Thanks for your comments. There is still a significant proportion of patients who for several reasons require pharmaceutical therapy of the underlying diseases. IN administration may overcome inherent limitations of acute oral administration of drugs, such as nifedipine which was removed due to its detrimental rapid blood pressure decline. There is an increasing proportion of patients with recurrent episodes of AF or angina despite the first line therapies (ablation or revascularization, respectively) who require non-invasive pharmaceutical therapy due to personal preferences, co-morbidities, age. Moreover, acute events may be more effectively treated with rapid administration of medications even out of hospital.
- in the light of the above, it is worth thinking about changing the title... it is difficult to call it a "gold player"...
We have changes in “alternative route”.
- please provide a broader discussion of the mentioned endothelial dysfunction, primarily through the guanylate cyclase pathway.
Thank you for your suggestion. A more extensive analysis has been added in section 4.1.
I think the topic is interesting (therefore worth publishing). The work cannot mislead the reader and therefore requires expansion and thorough reconstruction of the chapters, taking into account current clinical recommendations.
Thank you for your positive comments that helped us to revise, supplement and strengthen our work.
Reviewer 2 Report
Comments and Suggestions for Authors
The review by Papakyriakopoulou et al. discusses the use of intranasal (IN) administration for rapid systemic drug absorption, focusing on its potential in managing acute cardiovascular events. It emphasizes the advantages of IN delivery, such as bypassing first-pass metabolism and providing a swift onset of action. The review evaluates studies on IN formulations for conditions like angina and cardiac arrhythmias, highlighting promising results in preclinical evaluations and a limited number of clinical trials. Challenges include the need for more extensive clinical trials, but the study suggests that the nose-to-heart approach has clinical potential in cardiovascular medicine.
Considering that an acute cardiac event is a highly critical emergency situation, this thorough review is valuable for offering an insightful overview of the nose-to-heart approach. The goal is to expedite the rescue of patients more effectively. Although this approach is primarily discussed for its beneficial effects, it is essential to address the major drawbacks and potential side effects associated with administering medication through the nose.
Comments on the Quality of English Languageminor
Reviewer 3 Report
Comments and Suggestions for Authors
Overall, the review is informative and covers major papers published to date. However, some potentially important issues of nasal delivery of drugs should be discussed. Please see below for specific comments.
1. The review discusses as though drug absorption takes place within the nasal cavity. This may not be true. Drugs may bypass the nose cavity and reach the lungs where they may be taken into the blood and lymph. This route/possibility should be discussed, especially if drugs are encapsulated because the nasal mucus layer is the first defense for incoming foreign particles.
2. Nasal spray requires that drugs are in solution. In general, solubilized molecules, compared to the undissolved form, are much less stable. Stability (i.e. half life) and conditions of storage should be discussed.
3. Tables should include a column (or include in the disease column) indicating the subjects used (i.e. animal species, human-age, -disease, etc.).
4. Number the references.
5. If any of the images in Fig. 1 is a direct copy of published image/ diagram, identify the source.
Comments on the Quality of English LanguageMinor editing is necessary such as the style of citations in the text.
Author Response
Reviewer 3
Overall, the review is informative and covers major papers published to date. However, some potentially important issues of nasal delivery of drugs should be discussed. Please see below for specific comments.
- The review discusses as though drug absorption takes place within the nasal cavity. This may not be true. Drugs may bypass the nose cavity and reach the lungs where they may be taken into the blood and lymph. This route/possibility should be discussed, especially if drugs are encapsulated because the nasal mucus layer is the first defense for incoming foreign particles.
- Nasal spray requires that drugs are in solution. In general, solubilized molecules, compared to the undissolved form, are much less stable. Stability (i.e. half life) and conditions of storage should be discussed.
Thank you for your suggestions. The issues raised in comments 1 and 2 are discussed in the added section: 5. Disadvantages of IN drug delivery.
- Tables should include a column (or include in the disease column) indicating the subjects used (i.e. animal species, human-age, -disease, etc.).
Thank you for your comment. A relevant column has been added to the first table to define the formulation and in vivo preclinical studies, while the second table included only clinical studies as mentioned in the caption.
- Number the references.
All the references have been numbered and the commented studies are listed in the tables by year.
- If any of the images in Fig. 1 is a direct copy of published image/ diagram, identify the source.
The figure has been created by the authors on biorender.com. The citation if the site has been added in the caption.
Round 2
Reviewer 1 Report
Comments and Suggestions for Authors
The authors introduced appropriate corrections to the manuscript. I believe it may be considered for publication but the role of the GC/NO/CO pathway is greater (especially in terms of the range of diseases, hypertension, heart failure, pulmonary hypertension and many others), hence I would suggest changing the reference to a more recent one from recent years, not from 2007. The view presented therein will certainly be broader
Author Response
Following reviewer's suggestion we have changed reference 50 with the a more recent one.
Reviewer 3 Report
Comments and Suggestions for Authors
Authors fully responded to my comments. Ihave no further comments.
Comments on the Quality of English LanguageClear and reads well.
Author Response
We have corrected some typos. There are no further comments.